# Uncovering a New Concept of Foodnality in Diet Recommendations for Chinese Aging Populations Based on Systematic and Bibliometric Review

**DOI:** 10.3390/foods13244062

**Published:** 2024-12-16

**Authors:** Guanya Zhang, Junqiao Wang, Xiaojun Huang, Xiaoli Xu

**Affiliations:** State Key Laboratory of Food Science and Technology, China-Canada Joint Lab of Food Science and Technology (Nanchang), Key Laboratory of Bioactive Polysaccharides of Jiangxi Province, Nanchang University, Nanchang 330047, China; gyzhang@email.ncu.edu.cn (G.Z.); huangxiaojun0617@163.com (X.H.); xiaolixu78@sina.com (X.X.)

**Keywords:** foodnality, aging population, chronic diseases, dietary patterns, diet recommendations

## Abstract

China is grappling with significant aging challenge, and food patterns play an important role in the health of aging populations. A comprehensive bibliometric analysis with regard to systematically examined population-based studies published between 1 June 2014 and 31 May 2024, and 83 eligible studies, including 43 cross-sectional, 34 longitudinal, and 6 case–control studies, was conducted to investigate the correlations among foods, dietary patterns, and healthy aging concepts. An interesting conclusion from cohort studies was that foods recommended for intake in certain diets might not be recommended in other diets, i.e., food items in various diets showed different contributions. This dual character of foods exhibited in the diet is termed foodnality, a new concept proposed in this study. Foodnality was partially related to the origins, quality, and processing of foods, as well as individual physical status. Therefore, recommended foods with positive foodnality and unrecommended foods with negative foodnality were summarized, and four-dimensional diet recommendations were further proposed, including choosing a suitable dietary pattern, evaluating positive foodnality, upgrading foodnality to a premium diet, and cultivating healthier dietary concepts. This review plays a pivotal role in evaluating the guidelines on food intake and dietary patterns among elderly residents in China.

## 1. Introduction

Aging is an irreversible trend globally, spreading from developed countries to developing countries [1]. According to World Health Organization (WHO), China currently has the largest elderly population in the world, projected to reach 402 million residents aged 60 and over by 2040, accounting for 28% of the total population [2]. However, China first went through an instance of negative population growth, with 9.56 million births and 10.41 million deaths in 2022, experiencing an unexpected decline in the total population, and the population is projected to shrink by nearly half by 2100 [3,4].

A recent report in the *Lancet* revealed an increase in the loss of healthy life expectancy from 8.6 to 10 years between 2000 and 2019 [5]. Chronic diseases are one of the major threats to the Chinese healthy aging population, and approximately 190 million elderly people suffered from chronic diseases in 2021 [6]. Among them, 16% were partially disabled and 4.8% were completely disabled, with healthy life expectancy decreased from 68.7 years old to 68.4 years old [7]. The promotion of healthy aging was implemented as a key national policy within the ‘14th Five-Year Plan’, with the aim of maintaining social and economic stability [8]. In this context, diet plays a key role in nutritional support, chronic disease management, and the promotion of healthy ageing. This has prompted numerous studies on the relationship between diet and general health indicators through population monitoring [9].

As shown in Figure 1, diet can affect human health through diverse pathways. In the short term, ingested foods pass through the digestive system, including the mouth, esophagus, stomach, and duodenum, to transfer abundant nutritional components and stimulate the skeletal and smooth muscles of the digestive organs [10]. Small molecules are assimilated by cells, whereas larger biomolecules need further breakdown by the digestive system. Concurrently, foods engage in signaling, leading to the secretions of various glands and digestive enzymes involved in both anabolic and catabolic processes, transforming foods into nutrients for cellular delivery. On the other hand, diet can influence human metabolic capacity, gut microbiota, emotions, memory, fitness levels, immune responses, etc., over the long term [11,12]. Currently, several dietary systems and guidelines have been developed in China, such as the Chinese Healthy Diet Index, Chinese Dietary Guidelines Index, Chinese Food Pagoda, and Diet Quality Divergence Index [13]. International classical dietary patterns, such as the Mediterranean, Dietary Approaches to Stop Hypertension (DASH), and vegetarian patterns, have demonstrated the relationship between diet and health [14,15]. The Mediterranean dietary pattern emphasizes a good food source of monounsaturated fatty acids (MUFA) in olive oil, accompanied with intake of vegetables, nuts, seeds, fruits, and whole grains and reductions in the consumption of red meat and unhealthy fats. In the DASH dietary pattern, the main food items are low-fat dairy and fiber-based foods, and the western dietary pattern contains more red and processed meat, butter, French fries, refined grains, deserts, potatoes, sweets, high-fat dairy, and beverages. However, those dietary patterns may not be suitable for the Chinese aging populations because of specific cultural and metabolic differences. For instance, an isocaloric-restricted dietary pattern suggested as a preventive strategy to obesity and prediabetes might have resulted in higher risk of hypoglycemia [16]. Dietary patterns that emphasized adequate fruit intake were positively associated with hypertension and systolic and diastolic blood pressure but could cause negative prognostic effects on elderly type 2 diabetes mellitus patients [17]. These facts indicate that there are some limitations when making diet recommendations if dietary patterns are relied on without consideration of the complexity of food items, which may have dual effects on human health. For example, red meat contributed to nutrient supplementation and higher protein and muscle quality, but high intake of this food increased the risk of chronic metabolic diseases [18].

Therefore, in this study, we aimed to summarize the complex characteristics of food items in current dietary patterns by using both systematic and bibliometric analysis of the previous literature. In addition, food consumption was recorded to figure out the relationship between food attributes and chronic diseases among the Chinese aging population, as well as to develop a simplified recommendation guideline for the elderly.

## 2. Materials and Methods

This systematic and bibliometric review followed the PRISMA (Preferred Reporting Items for Systematic Reviews and Meta-Analyses) 2020 guidelines [19]. In the process, the PRISMA procedure was used to select relevant articles, as summarized in Figure 2. Scientific mapping techniques, such as coword and co-occurrence analysis, were then applied to identify exposed foods and dietary patterns, chronic disease outcomes, and dietary recommendations.

### 2.1. Population and Chronic Disease Definitions

This study began with studies targeting individuals with chronic diseases, and the population definition was based the latest China Health and Retirement Longitudinal Study (CHARLS), the nationally representative survey for the middle-aged and elderly population in China with ages of 45 and above [20]. Self-reported information concluding chronic diseases from CHARLS was analyzed for the health indicators of the aging population in China. Six categories of diseases and related symptoms were screened, which also encompassed the reference indicators for healthy aging defined by WHO, and the six categories included (1) cardiovascular disease (CVD): hypertension, heart attack, dyslipidemia, stroke; (2) metabolic syndrome (MetS): obesity, diabetes, liver disease, kidney disease, stomach disease, gastrointestinal disease; (3) cognitive impairment and mental illness (CIMI): depression, memory-related disease, mild cognitive impairment (MCI), Alzheimer’s disease; (4) cancer (CA): different types of cancer; (5) physical quality (PSQ): arthritis, frailty, bone health, fracture, all-cause mortality; (6) respiratory disease (RD): chronic lung diseases and asthma. However, no articles related to RD met the requirements after the literature review, so CVD, MetS, CIMI, CA, and PSQ were finally designated as chronic diseases in this review.

### 2.2. Data Source and Screening

Searches on Web of Science (WOS) comprising all databases, including WOS core databases, BIOSIS Citation Index, Chinese Science Citation Database, Korean Journal Database, MEDLINE, Preprint Citation Index, SciELO Citation Index, were conducted to search studies published between 1 June 2014 and 31 May 2024. We used ‘diet’ as the topic word, abstract word, or author keyword in the search strategies separately, and used Medical Subject Headings (MeSH) including ‘China’, ‘aged’, ‘middle aged’, ‘aging’, and ‘aged 80 and over’ for further screening. In addition, further citation tracing searches were performed to ensure research comprehensiveness to retrieve studies missed by the predefined searches.

### 2.3. Study Selection

Food items, food groups, dietary patterns, and diet recommendations were evaluated without limitations or restrictions imposed on the geographical location, gender, lifestyle or income status of participants. Abstracts, protocols, conclusions, and full texts were previewed to exclude studies that did not meet the criteria: (1) were of cross-sectional cohort, longitudinal cohort, or nested case–control design, including follow-up studies of randomized controlled trials; (2) had participants aged ≥45; (3) provided recommended or unrecommended conclusions on the associations between chronic diseases and food items, food groups, or dietary patterns. When exposure was dietary patterns, food items of factor loading ≥0.2 were screened; (4) specific food items were contained (studies targeting micro- and macronutrients were excluded).

### 2.4. Data Analysis

The COOC 13.5/VOSviewer 1.6.13 software was used for data selection and association, and duplicated results were removed by COOC. The disease incidence and the corresponding frequencies of recommended and unrecommended foods in regard to associations between food items and chronic diseases (CVD, MetS, CIMI, CA, and PSQ) were statistically analyzed by clustering analysis methods. Finally, dietary strategies beneficial to the aging population were summarized and proposed.

## 3. Results

### 3.1. Literature Overview

This review chose single sources to mitigate possible errors derived from bibliometric software and keep consistent the issued timing within 10 years from the database. From the original search in the WOS database (Figure 2, *n* = 1728 citations), 332 articles were screened using defined CHARLS filter criteria, and 92 articles were selected after exclusion. Further scrutiny of the full texts excluded eight articles and one additional study that showed no association between food consumption and chronic diseases. Consequently, a total of 83 articles were collected with all inclusion criteria, including 43 cross-sectional, 34 longitudinal (with follow-up periods ranging between 1.0 and 15.8 years), and 6 case–control studies (with follow-up periods ranging between 0.25 and 5.0 years). The study designs, participant characteristics, specific ailments or physical status, chronic diseases, exposure and outcome assessment, exposure of food groups or patterns, and main conclusions are summarized in Table 1. Among the selected studies, 42 cohort studies were conducted nationally, with sample sizes ranging between 1086 and 59,980; 26 cohort studies were provincial, with sample sizes ranging between 276 and 30,484; and 15 cohort studies were at the municipal level, with sample sizes ranging between 210 and 5281. The ages of participants ranged from 50.13 ± 4.81 to 92.9 ± 7.5 years. Food frequency questionnaires (FFQs) were used for data collection in most cohorts, other assessments included 24 h dietary recalls for 3 consecutive days, 3-day dietary records, and self-reports. Plant-based, vegetable-based, and traditional Chinese dietary patterns were common dietary patterns among local residents, and only eight studies involved western classical dietary patterns such as the Mediterranean and DASH diets [21,22,23,24,25,26,27,28].

### 3.2. Classification of Food Items

Clustering analysis were conducted to classify the food items (Figure 3). Among them, five food groups showed positive intervention, including (1) vegetables, vegetable oils, tubers, teas, whole grains, soybean products, miscellaneous beans, and honey; (2) fruits, legumes, low-fat dairy products, nuts, soy, whole grains, and vegetables; (3) algae, bean products, eggs, fish, garlic, meat, seafood, sugar, and eggs; (4) aquatic products, coarse grains, dairy products, fresh fruits, fresh vegetables, legume products, milk, mushrooms, seeds, snacks, and legumes; (5) dairy, pork, poultry, red meat, fish, and rice. On the other hand, five groups of food items showed negative interventions, including (6) red meat, alcohol, animal fats, aquatic products, candies, fried foods, meat, sugars, and refined grains; (7) alcoholic beverages, coffee, eggs, organs, poultry, processed meat, seafood, snacks, and red meat; (8) pork salt, salt-preserved vegetables (SPVs), tubers, wheat, and cereals; (9) fats, fish, shrimp, and rice; (10) cakes, dairy products, and fast foods.

### 3.3. Food Items in Association with Chronic Disease Symptoms

Food items were then categorized based on their statistical likelihood of being recommended or unrecommended for elderly individuals with physical status. In most cases, vegetables were regarded as recommended food items and showed positive effect to decrease chronic disease incidence. Fruits followed behind, also being recommended except for elderly individuals with MetS. Fish and nuts had positive correlations with CIMI, CVD, and PSQ, and animal-origin foods, such as animal meat and organs, displayed higher associations with chronic diseases in the Chinese elderly population. After clustering analysis of food items and specific chronic disease symptoms, their potential associations were summarized as shown in Figure 4.

CVD. A habitual meat diet, combined with consumption of modern and processed foods (fried, refined), in the aging population was associated with increased hypertension and atherosclerosis, which could lead to CVD or equivalent risk [30,31,33,34,36]. One 4-year longitudinal cohort warned that 50 g/day or higher red meat consumption was significantly associated with increased incident cardiocerebrovascular diseases (CCVDs) (aHR = 1.10) among high-income subjects and urban residents [35]. Alternately, complex carbohydrates and unsaturated fats from food items such as whole grains, nuts, and fish were advocated to avert high blood pressure [21,29,32]. Traditional dietary patterns and whole plant diets containing vegetables and fruits as the basic foods helped to keep vascular health among Chinese aging populations [30]. On the contrary, western dietary patterns containing food items such as red meat, sweets, and sugar beverages correlated with blood pressure variability in the elderly.

MetS. Industrialization and urbanization caused changes in the dietary structure of the Chinese elderly population and led to increasing trends in the consumption of foods such as meat, fat, vegetable oil, dairy, etc. The calories from foods Chinese elderly individuals ate exceeded energy expenditure, so as to cause problems of overweight and obesity [35,37,38,40,48]. Lower intake of bean products was associated with gastrointestinal discomfort among Chinese urban elders [46]. Processed animal meat was an inducing factor associated with metabolic syndrome diseases such as liver metabolism, hyperuricemia, prediabetes, and diabetes. Therefore, decreased intake of a western diet pattern containing high proportions of animal meat, dairy, and fried food may be essential to the early prevention of MetS among the Chinese aging population [39,41,42,43,45,47,51,53]. Carbohydrate calories acquired from staple foods such as refined grains, commonly found in each meal of a Chinese family, easily exceeded energy intake. For example, one 2-year longitudinal cohort study showed that higher consumption of 250–500 g staple food was significantly associated with higher risk of obesity, with an odds ratio (95% CI) of 1.26 (1.07–1.48) [50]. A cross-sectional cohort study showed that higher consumption of a lactoovovegetarian dietary pattern, a balanced diet with whole grains, vegetables, eggs, and dairy products, significantly lowered sarcopenic obesity incidence, with an odds ratio (95% CI) of 0.79 (0.65–0.97) [49]. On the other hand, long-term tea consumption and diverse food patterns were independent protective factors for decreasing the risk of metabolic syndrome [48,52].

CA. Overprocessing, such as high cooking temperature and long cooking period, can be detrimental to food attributes [11]. One 11.1-year provincial longitudinal study used International Classification of Disease version 10 codes to access the outcome of cancer mortality and found that fried food, processed vegetables, and baked cereal products increased exposure to harmful compounds such as acrylamide and thus increased the risk of cancer [54]. Another 6.9-year national longitudinal study was conducted among 63,257 Singapore Chinese participants with outcome assessment coming from the Singapore Cancer Registry, and the result indicated that meat–dim sum consumption was associated with higher possibility in the incidence of gastric adenocarcinoma compared with a vegetable–fruit–soy diet [55].

CIMI. Thirty-two studies investigated cognitive impairment and mental illness, and the main exposures of dietary patterns were in the traditional western pattern [61,66], plant-based pattern [72,77,78,79,83], vegetable–fruit pattern [73,74,80] and dietary diversity score pattern [76,81]. High-quality diets were rich in fiber, premium protein, and healthy fat, which benefited aging populations from midlife throughout late life effectively in lowering the incidence of MCI [23,58,70,71,72,73,74,75,76,77,78,79]. Multiple studies showed that high-quality foods such as nuts, fish or aquatic products, and legumes were beneficial for CIMI [22,24,56,59,60,65,66,70,71,77,78,81,82]. For the Chinese aging population, steamed vegetables, tea oil or linseed oil, and freshwater fish in the diet instead of the salads, olive oil, and ocean fish of Mediterranean diet were suggested because of their easy accessibility [22,56,77,83]. Certain foods, such as meat, offal, processed food, and fried food with VB-oil, might cause negative effects on cognitive function [61,62,64,69,77,78,79]. A cross-sectional study used the Montreal Cognitive Assessment to assess cognitive impairment and found that less than 20 g/day consumption of cooking oil decreased the risk of disease and increased cognitive function [56]. A healthy lifestyle, including consuming positive food items frequently, developing good dietary habits, and reducing alcohol and tobacco consumption, can help to prevent senile dementia and cognitive impairment among the Chinese aging population [57,63,68,71,75,85]. The economy was part of what supported the elderly in consuming high-quality foods with fresh, clean, and healthy characteristics and maintaining good mental health [58,60,67].

PSQ. Twenty-three studies investigated the influence of food terms and dietary patterns on physical quality, including all-cause mortality, frailty, anemia, hip fractures, sarcopenia, and bone mineral density. The major dietary patterns included the Healthy Diet Indicator score pattern [26,27], vegetable-based pattern [85,103], dietary diversity score pattern [92,94], and plant-based pattern [101,102]. Chinese elderly individuals prioritized food characterized by controlled calories, such as fresh vegetables and fruits, and those helped to improve movement ability and decrease the risk of mortality [26,85,86,87,92,94,95,97,102]. Eating habits influence individual health, and consuming foods with different types, even some with negative properties, was healthier than choosing only certain foods [92,94]. With decreased digestion ability, hormone secretion, and muscle weight, milk or formula milk powder, eggs, and fish were necessary for active supplementation of protein and calcium [28,88,89,94,97,98]. Fish was rich in high-quality fat as well, so higher intake of fish came together with more premium dietary oil consumption, indicating that nutrient categories could serve as a measurement for the evaluation of food attributes [100]. Cohort studies indicated that the consumption of vegetables, whole grains, and fungi were positive for preventing sarcopenia, fracture, and frailty, as dietary fiber improved protein absorption rates for gaining more muscle [28,90,93,99]. Western-style patterns could improve the mass of skeletal muscle; however, Chinese aging populations were encouraged to consume more plant-based foods to keep overall fitness [26,91,96].

### 3.4. Positive and Negative Foodnality

The relationship between food and the health of aging populations was so complicated that certain food items sometimes could exhibit contrary effects in different chronic diseases or physical statuses. Food contributions also varied in different dietary patterns, individual characteristics, and processing and storage methods. The systematic and bibliometric analysis delved into this intricacy, grounding conclusions in population studies that considered the diverse food consumption of daily life. These food attributes, which can be likened to the ‘personality’ of foods, should be considered when giving diet recommendations. Therefore, we created a vivid term of ‘foodnality’ to emphasize the dual character of foods. If food items were recommended for intake, we defined their foodnality as positive, and unrecommended items were regarded as negative. In total, 101 positive and 92 negative food items were documented as related to chronic diseases or individual physical status and foodnality (Appendix A). As shown in Figure 5, the top 10 food items with positive foodnality, along with their corresponding frequency of appearance in recommendations, were vegetables (50), fruits (48), fish (31), nuts (29), whole grains (18), tea (16), legumes (15), eggs (14), tubers (11), and mushrooms (11). Among them, only fish and eggs were animal-derived, and all the other positive food items originated from plants. On the other hand, the top 12 food items with negative foodnality included red meat (17), processed meat (13), meat (11), poultry (9), fast foods (8), dairy products (8), eggs (7), sugars (7), fish (6), wheat (6), alcohol (6), and refined grains (6), and all of them either were of animal origin or underwent a refining process before consumption (Appendix A).

### 3.5. An Integrative Theoretical Framework Through Thematic Analysis

The optimized diet recommendations, a well-balanced dietary pattern composed of positive-foodnality items with sufficient nutrients, were then concluded. As shown in Figure 6, four brief dimensions of diet guidelines for the elderly were created to keep them healthy. The first dimension was dietary patterns, and five recommended dietary patterns for Chinese aging populations were summarized, i.e., a plant-based pattern [26,29,77,79,83,85,101,102], a grain–vegetable pattern [39,42,47,61,66], a traditional Chinese pattern [31,34,37,40,45], a vegetable–fruit pattern [55,73,74,80], and an anti-inflammatory pattern [82,90,99]. The second dimension was positive foodnality with consideration of physical functions [27,32,35,43,46,51,59,70,96,100], i.e., foods items with positive foodnality [23,30,36,48,57,62,84,86] and high quality [25,28,44]. The third dimension was a well-balanced and healthy diet, suggested to adjust the dietary patterns based on foodnality. This recommended following a balanced diet [41,49,91], rebuilding a healthy diet structure [67,71,97], or combining plant-based and meat-origin foods [64,78] as well as traditional and western foods [38,88]. The fourth dimension was dietary concept, i.e., encouragement of diverse diet [52,76,81,92,94,95,98], control of overall food intake [50,58,68,75,87,93], paying attention to nutrient supplementation [63,89], and being aware of dietary consciousness [53,60] and healthy culinary methods [54] for the aging population.

## 4. Discussion and Conclusions

Population-based longitudinal, cross-sectional and case–control cohort studies were reviewed to investigate the associations between foods and the possibility of chronic diseases, as well as their symptoms, among the Chinese aging population. Understanding the complexities of diet and their impacts on chronic diseases leads to a realization that labeling foods simply as good or bad is not appropriate. From the current cohort studies, foods displayed both advantages and disadvantages in different origins, qualities, processing methods, and consumption scenarios. For example, milk is a highly popular food in western diets, while it is not as friendly for mostly lactose-intolerant elderly Chinese individuals. Therefore, the term ‘foodnality’ was defined with consideration of foods’ dual character, and it is suggested to make diet recommendations based on the foodnality of food items instead of strictly following the dietary patterns. In this study, three different levels support aging consumers to better understand foodnality and the related guidelines for choosing a healthier diet.

First, foodnality is influenced by many internal and external factors, such as food characteristics, nutrients, regional impact, processing or culinary methods, dietary intake, etc. Nutritional composition is the most fundamental factor of food, and basic nutrients such as protein, fat, carbohydrates, and trace elements are closely related to the health of the elderly. For instance, Chinese Food Pagoda advocates for older adults to increase their protein intake, with a recommended range of 1.2–1.5 g/kg body weight to prevent muscle degradation. However, the objective fact is that the protein intake among elderly people is far from reaching this standard, even in European regions where animal meat diets are predominant [104]. It is worth noting that excessive nutrients might bring a burden to the elderly, so food items should be selected according to the individual’s status. Gender-specific responses to diet found that men might be more prone to weight gain than women by consuming staple foods like rice and wheat, and postmenopausal women might show greater sensitivity to dietary changes due to hormonal fluctuations [29,105]. In addition to differences in physical condition caused by gender, elderly women were more likely to benefit from consuming vegetables and fruits. One reason for this could be that elderly women were more knowledgeable about healthy food and dietary conception [106]. Geography influences physical status, dietary concept, and availability of food resources between coastal and inland regions; this affects foodnality as well. In China, regional dietary preferences vary; for instance, rice, pork, and legumes were more prevalent in Southern regions, whereas dairy, wheat, and other grains were more widely consumed in the North [33,52], and this resulted in variation in the nutritional composition of foods [107]. Even for foods in the same area, there might be variations in nutritional composition and physiological functions [108]. Processing and cooking methods for foods involved in the food supply chain, such as the agricultural and food industries, the food service sector, and family kitchens, change the nutrients and quality of foods [109,110,111], thereby inevitably changing the foodnality. Overprocessing probably causes the degradation of nutrition and is especially detrimental for heat-, oxygen-, and salt-sensitive nutrients, resulting in negative foodnality in the eligible database [23,85,97]. Fried, salted, stale (overnight, contaminated, transported, and repeatedly processed) foods even induced food safety issues [11]. Salt-preserved vegetables were not recommended to be consumed by aging people who had MetS, CIMI, or PSQ [52,83,87,97]. Even minimal processing could substantially reduce the nutrient density, as evidenced by the lower nutritional profiles of dried and fresh vegetables and fruits [112].

Foodnality is also influenced how much food an individual intakes, especially for aging populations. For example, compared with no or less-than-weekly consumption of eggs, infrequent low-quantity consumption (1–2 days/week and 0.1–1.9 eggs/day), infrequent high-quantity consumption (1–2 days/week and ≥2.0 eggs/day), and frequent low-quantity consumption (≥3 days/week and 0.1–1.9 eggs/day) were associated with lower risk of cognitive impairment, but no association was found among frequent high-quantity consumers (≥3 days/week and ≥2 eggs/day) [75]. The variable about dietary habits may contribute to the inconsistency between egg and fish consumption and their possibilities in chronic disease such as MetS in some cohort studies [39,45].

Second, since foodnality is closely correlated with the occurrence of chronic diseases to some extent, healthy dietary concept related to positive foodnality is highly recommended for the elderly. Vegetables and fruits have shown beneficial effects on most symptoms besides physical strength, and premium protein and mineral sources such as fish, milk, eggs, and legumes are recommended for joint consumption [25,26,28,97]. Red meat and poultry should not be prioritized, as they showed high-frequency correlation with chronic diseases symptoms in elderly Chinese individuals [42,43,45,47,53]. Pork and soybean are relatively neutral food factors and can be consumed as protein sources several times per week. Attention should be paid on high-caloric diets to problems such as eating too much saturated oil, which is commonly found in foods distributed in catering, snack, retail, and distribution markets. For the aging population, high-cost unsaturated fatty acids are less approachable. Therefore, a dual-pronged strategy should be recommended to minimize the intake of VB-oil and optimize fat options for the elderly in ordinary economic conditions, such as increased consumption of aquatic products (mainly fish) and nuts and decreased intake of fried food, animal fat, and meats. Food with negative foodnality in diet should be replaced or at least reduced, benefitting health [26]. Carbohydrates, omnipresent in traditional Chinese meals, can quickly surpass daily recommendations when coupled with snacks or sweet desserts. It is important to guide aging people to realize that an optimized ‘foodnality’ might involve incorporating more fiber, such as whole grains, legumes, algae, and fungi such as mushrooms, into the daily diet. The elderly also need to be conscious of excessive consumption of sugar and salt. To better improve the dietary concept of aging populations, popularization of science programs focused on food selection and family cooking aimed especially at preventing CVD and MetS are recommended. Given current nutritional knowledge, educating elder adults on balanced diets is necessary, and analyzing diet-related chronic disease symptoms is essential to judge foodnality as well.

Third, diet recommendations based on foodnality can be made for aging populations. Basically, consumers should evaluate the importance of the quality, diversity, and collocation related to foods. Theoretically, the priority is targeting high-quality food items, namely fresh vegetables, fruits, fish, nuts, whole grains, and foods with rich nutrients and simplified processing, for elderly people [25,28,44]. Increasing dietary diversity can ensure the intake of various foods to promote health in regard to CIMI and PSQ, creating benefits for both the physical and mental well-being of the elderly [76,81,94,95]. In the CLHLS cohort study, the dietary diversity score was calculated according to the intake frequency of 13 food groups, and the low diversity score was defined as <7 [92]. It is suggested to consume diverse foods as much as possible to improve frailty, even together with some foods with negative foodnality. In addition, food items have coordinating or resistance effects in a diet, so elderly people are recommended to screen food items according to their own physical status. For example, a traditional Chinese plant-based diet characterized by more intake of nuts, seeds, tubers, vegetables, legumes, and fruits was beneficial for obesity. However, this diet might display negative effects on anemia, suggesting that professional health assessment and dietary pattern adjustments are necessary when judging foodnality [69]. In modern dietary patterns with higher carbohydrate and animal meat proportions, consuming fruits could become burdensome for aging consumers; similar findings were noted when fruits were consumed with seafood [16,40].

Therefore, five positive food groups and five negative groups were concluded in this review. An intriguing trend is that while positive food groups contained rich, high-frequency positive foods such as fruits, vegetables, and nuts, they may also include some high-frequency negative foods such as red meat and poultry, implying there still exists room to optimize foodnality to achieve a premium diet. Even some international popular dietary patterns, including the Mediterranean, DASH, pescetarian, vegetarian, and Paleolithic patterns, cannot apply to all populations [46,113,114], given that differences exist in the major food items and culinary methods used between Chinese and western cuisines. For example, the Mediterranean diet emphasizes whole grains and olive oil as sources of energy, but it is more reasonable to suggest Chinese aging individuals to consume more coarse cereals, tubers, tea oil, or linseed oil in order to enjoy similar positive foodnality. In the DASH diet, the high intake of fruits and vegetables aligns with the plant-based foods in traditional Chinese diets; however, milk is not suitable for Chinese individuals with lactose intolerance. Chinese aging individuals may need to seek lactose-free dairy products to shift the foodnality in the DASH diet to enhance the dietary quality. The absence of fish in vegetarianism may not suit Chinese elderly individuals who require sources of high-quality protein; therefore, it is recommended that vegetarian seniors undergo dietary assessments and supplement with some animal foods with positive foodnality. Considering all these variations in food items and groups, dietary patterns significantly vary in overall nutrients composition and exposure outcome.

In conclusion, foodnality exists, and it is related to the characteristics and consumption scenarios of foods. Foodnality is important for aging populations to better understand the dual character of foods based on the quality and processing of the food as well as the individual physical status of the consumer. Therefore, understanding the concept of foodnality enables the elderly to achieve a quicker selection of suitable food items according to their actual situation.

### Limitations and Future Research

The findings revealed that foodnality is important in the association between diet and health. However, the interpretation of the new concept of ‘foodnality’ is extremely complex, since substantial variations existed across the studies in terms of the definitions of outcomes, measurement of exposure, cohort design, and sample size. There is limited research on educational strategies to help older adults apply the concept of foodnality in their daily lives, implying that the development of personalized and accessible guidelines for food choice based on food quality is necessary. Further research is needed on how to communicate the concept of foodnality in a clear and practical way to the older population so that older individuals can make informed food choices. In addition, future studies could focus on developing rapid foodnality assessment tools that allow consumers to easily compare the nutritional quality and potential effects of different foods.

## Figures and Tables

**Figure 1 foods-13-04062-f001:**
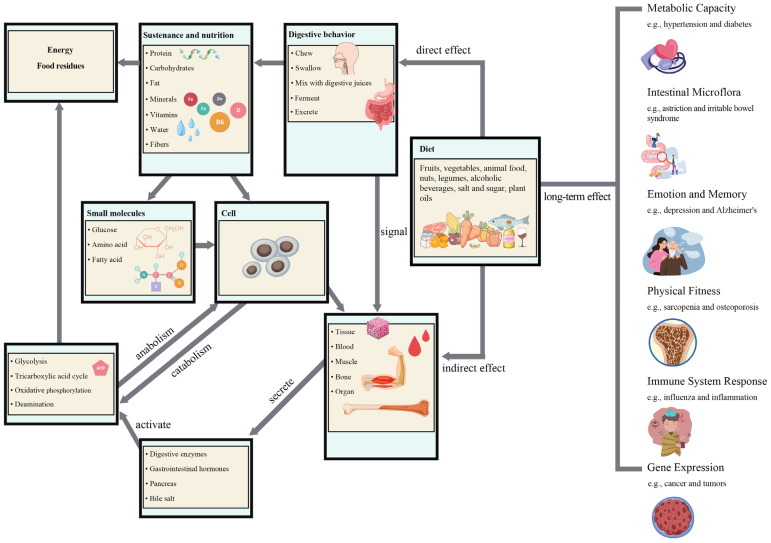
Interaction between diet and human health through three main pathways. Direct effect: food is the source of nutritional supply, providing energy, carbohydrates, fats, proteins, vitamins, minerals, etc. to regulate various biochemical processes. Indirect effect: biochemical and physiological responses. Food delivery signals to activate the function of the digestive system and biomolecules to participate in organism reactions such as synthetic metabolism and catabolism. Long-term effect: disease risk, a significant protective or harmful impact on health.

**Figure 2 foods-13-04062-f002:**
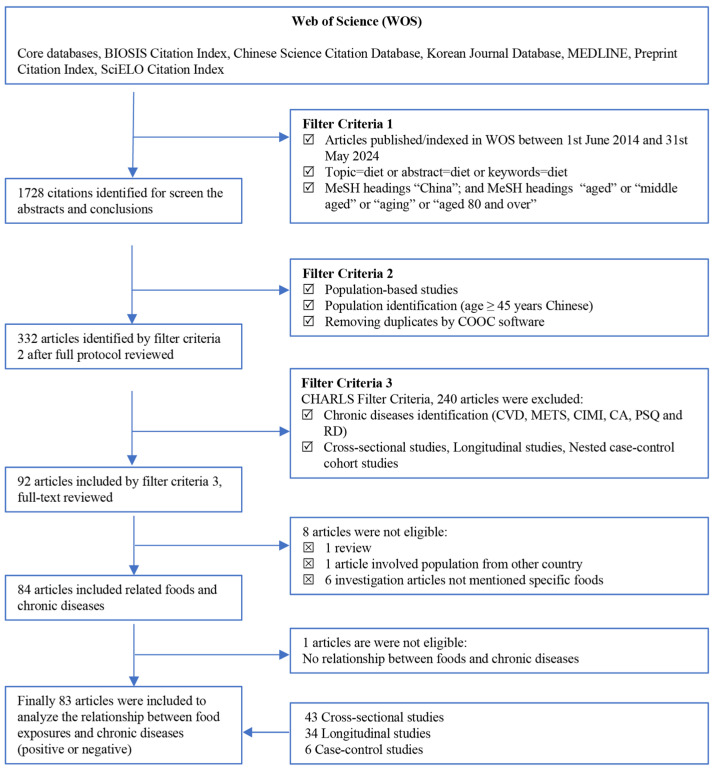
Flowchart selection for target population studies. This flowchart describes the databases, filter criteria, and searching processes used in the study. Points of screening were as follows. (a) The population study (clinical or questionnaires) included only Chinese people (including Singapore Chinese) aged 45 years old and above, excluding children, pregnant women, and lactating Chinese adults. (b) Data from longitudinal studies, cross-sectional studies, nested case–control studies, and randomized controlled trials were used. (c) Reviews, letters, meta-analyses, and editorials were excluded. (d) The selection of food matrices (nutrients or food groups) was based on the conclusions of studies or their obvious positive (recommended) or negative (unrecommended) impacts on chronic diseases. (e) Even if foods were included in the recommended or unrecommended groups, if they were not specifically highlighted, they were excluded. (f) Different categories were defined for the same food if there were changes in composition or chemistry, for example, fruit and dry fruit or meat and processed meat.

**Figure 3 foods-13-04062-f003:**
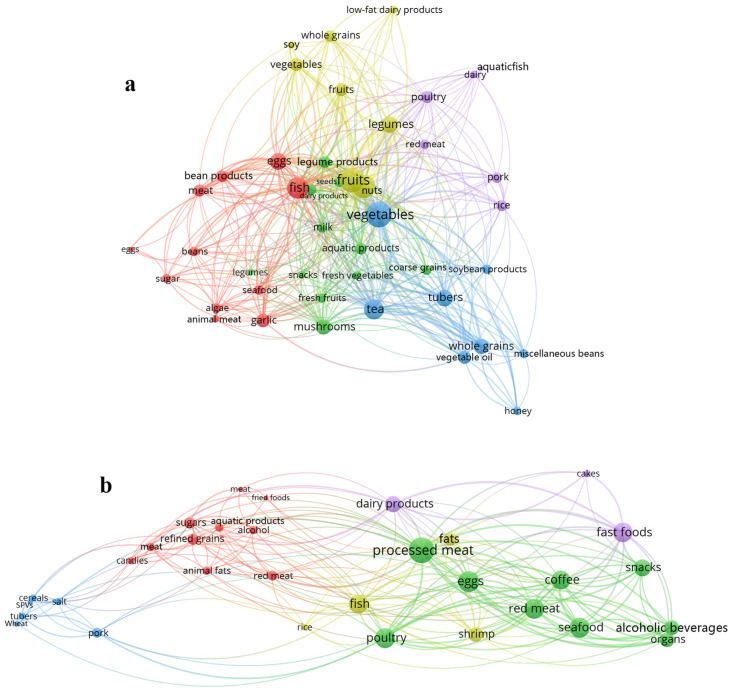
Clustering analysis of recommended food items (**a**) and unrecommended food items (**b**) based on co-occurrence matrix. Foods labeled with the same color were frequently found in similar food combinations and were associated with specific health conditions. The lines connecting the foods represent the degree of association between them, with thicker lines indicating stronger connections.

**Figure 4 foods-13-04062-f004:**
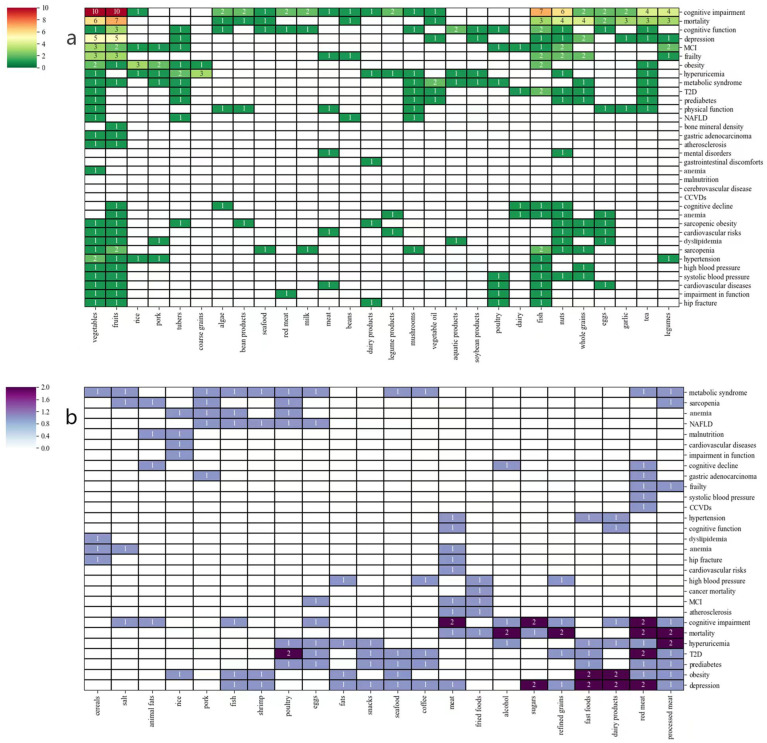
Clustering analysis of recommended food items (**a**) and unrecommended food items (**b**) in association with chronic disease symptoms based on bimodal matrix. The bimodal matrix diagram represented the relationship between food types (horizontal axis) and the physical functions of chronic diseases (vertical axis). Each cell in the matrix displayed a frequency number, reflecting the degree of association.

**Figure 5 foods-13-04062-f005:**
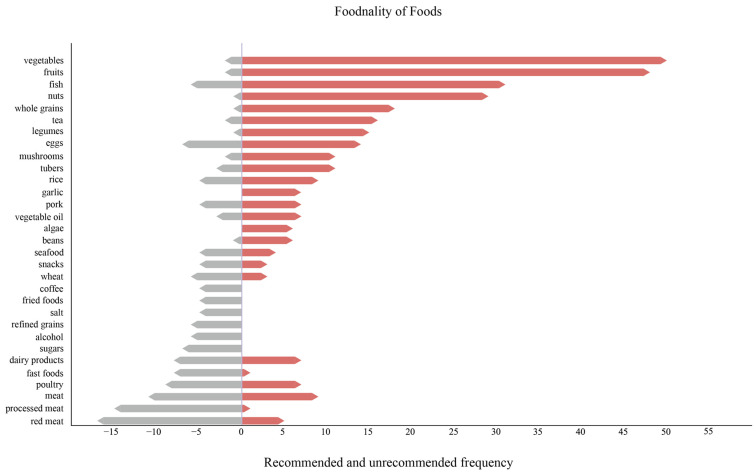
Frequency ranking of recommended and unrecommended foods. Red-colored pillars represent the recommended food items in cohort studies, namely those with positive foodnality. On the contrary, grey-colored pillars represent food items with negative foodnality. The length of each pillar represents the frequency of occurrence.

**Figure 6 foods-13-04062-f006:**
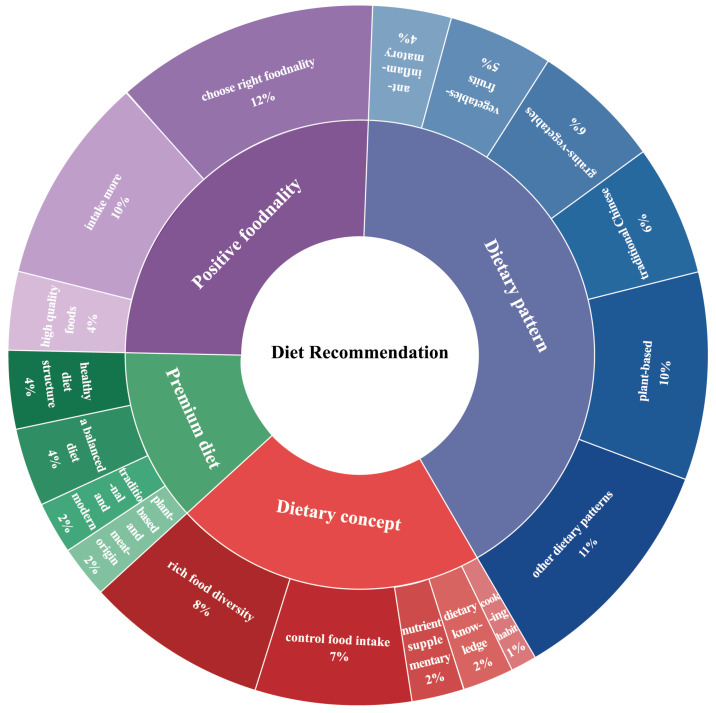
Four dimensions of diet guidelines with intuitive dietary advice. All advice was derived from the concise conclusions and frequency analysis after a full-text reading and generalization of each study. The percentage value represents frequency.

**Table 1 foods-13-04062-t001:** Characteristics of studies included in the systematic review.

Ref.	Study Design ^a^	Participants (Female)	Age Range ^b^	Physical Status	Outcome Assessment	Assessment of Exposure, N(n) ^c^	Food Patterns	Main Conclusions
Na, 2022 [21]	Longitudinal (City, 1)	324 (108)	≥60 (66.3 ± 6.0)	Systolic Blood Pressure (CVD)	Clinical and laboratory examination	Validated FFQ (40)	DASH diet	DASH diet pattern was associated with lower nighttime BP variability in elderly adults
Qin, 2015 [22]	Longitudinal (Nation, 5.3)	1650	55–65	Cognitive Decline (CIMI)	Linear mixed-effects models	24 h recalls over three days	Mediterranean, wheat-based, rice–Pork Patterns	An adapted Mediterranean diet or a wheat-based, diverse diet with similar components reduced the rate of cognitive decline
Tong, 2021 [23]	Longitudinal (Nation, 3)	14,683 (8674)	45–74 (72.9 ± 6.3)	Cognitive Impairment (CIMI)	Singapore- modified version of the Mini-Mental State Examination	Validated FFQ, 10 (165)	DASH diet	Improving diet quality from mid to late life was always associated with a lower risk of cognitive impairment in late adulthood
Huang, 2022 [24]	Cross-Sectional (Nation)	11,245 (6156)	≥50 (84.06 ± 11.46)	Cognitive impairment (CIMI)	The Mini-Mental State Examination	Validated FFQ, 12	cMIND diet pattern	Higher cMIND diet score was associated with better cognitive function
Zeng, 2014 [25]	Case–Control (Province, 4)	1452 (109)	≥55 Study (71.0 ± 7.0, *n* = 726), control (70.9 ± 6.9, *n* = 726)	Hip Fracture Risk (PSQ)	Anthropometric measurements, clinical assessment	Validated FFQ and diet-quality scores (79)	Healthy Eating Index-2005, the Alternate Healthy Eating Index, the Diet Quality Index-International, the Alternate Mediterranean Diet score	Avoiding intaking foods with low diet-quality scores was suggested
Neelakantan, 2018 [26]	Longitudinal (Nation, 16)	57,078	45–74	Mortality (PSQ)	International Classification of Diseases 9th and 10th Revision codes	Validated FFQ and 24 h dietary recalls (165)	Alternative Healthy Eating Index–2010, alternate Mediterranean diet, DASH diet, Healthy Diet Indicator score	The specific foods and dishes that characterized a high diet-quality score were different for Chinese and western populations
Zhu, 2018 [27]	Longitudinal (Province, 14.4)	30,484 (18,458)	70–86 (77.7 ± 4.74 for Male, 78.1 for Female)	Impairment in Function (PSQ)	Self-report	Validated FFQs and multiple 24 h dietary recalls recorded twice a month	Chinese Food Pagoda, DASH diet, the Alternative Healthy Eating Index Patterns	Higher consumption of fish, poultry, vegetables, and fruit and moderate consumption of red meat delayed the onset or development of aging-related impairments
Wang, 2021 [28]	Cross- Sectional (Province)	780 (445)	65–74 (66.85 ± 2.64)	Frailty (PSQ)	Five-item scales based on frailty phenotype	Questionnaire (27)	Protein-rich, vegetable, sugar–oil–condiment, DASH patterns	A better diet quality as characterized by DASH and ‘protein-rich’ was associated with a reduced prevalence of frailty
Liu, 2015 [29]	Cross- Sectional (Province)	726 (726)	48–70	High Blood Pressure (CVD)	Self-administered by validated and structured symptom checklist	Validated FFQ, 11 (85)	Processed food, whole plant food, animal food patterns	Adopting a whole plant foods diet rich in vegetables, fruits, and whole grains was suggested
WOO, 2018 [30]	Longitudinal (Province, 2)	4000 (2000)	≥65 (71.85 ± 4.74 for male, 72.01 ± 5.06 for female)	Atherosclerosis (CVD)	Laboratory measurements	FFQ, 32 (280)	Vegetable–fruit, snack–drink–milk product, meat–fish patterns	The importance of fruit and vegetable intake in any pattern was emphasized
Xu, 2018 [31]	Cross- Sectional (Nation)	2634 (1388)	≥60	Hypertension (CVD)	Clinical and laboratory examination	Three-day 24 h dietary recalls, 27	Traditional, modern patterns	Traditional dietary pattern (high intake of rice, pork, and vegetables) was significantly inversely associated with hypertension
Liu, 2021 [32]	Case–Control (City, 0.5)	210 (99)	≥60	Cardiovascular Diseases Indicators (CVD)	Hospital records	Validated FFQ and 24 h dietary recalls (95)	Not mentioned	Adherence to long-term walnut supplementation was good for CVD prevention
Song, 2021 [33]	Cross- Sectional (Nation)	3387 (1674)	≥60 (67.3 ± 5.9)	Dyslipidemia (CVD)	Anthropometric and clinical data	Three-day 24 h dietary recalls, 18	Balanced diet, western diet, thrifty diet Patterns	Balanced diet could reduce the risk of dyslipidemia
Li, 2022 [34]	Cross- Sectional (Province)	1136 (0)	≥65 (72.0 ± 9.0)	Cardiovascular Risk (CVD)	Hospital records	Validated FFQ, 10 (50)	Animal-based and processed food, traditional food, ovolactovegetarian food patterns	Choosing traditional and ovolactovegetarian diet patterns was suggested
Sun, 2023 [35]	Longitudinal (Nation, 4)	59,980 (31,606)	≥65 (69.68 ± 3.01)	CCVDs (CVD)	Clinical and laboratory examination	Validated FFQ	Not mentioned	50 g/day higher red meat consumption at baseline was significantly associated with increased incident CCVD among high-income subjects and urban residents
Wang, 2023 [36]	Longitudinal (Nation, 3.8)	9740 (5581)	≥65 (88 ± 11.4)	Hypertension (CVD)	Interview with family relatives	Simplified FFQ	Healthful plant-based diet index pattern	Increasing fruit and vegetable consumption was part of a healthy diet for the prevention of CVD
Shu, 2015 [37]	Cross- Sectional (Province)	2560 (1294)	45–60, Study (54.82, *n* = 453), Control (51.48, *n* = 1465)	Obesity (MetS)	Anthropometric measurements	Validated FFQ and the Kaiser–Meyer–Olkin measure	Animal food, traditional Chinese, western fast food, high-salt Patterns	The traditional Chinese pattern was associated with a lower risk of abdominal obesity
Xu, 2015 [38]	Cross- Sectional (Nation)	2745 (1445)	60–80	Obesity (MetS)	Anthropometric measurements	Three-Day 24 h dietary recalls (27)	Traditional, modern patterns	Combining traditional and modern patterns to reach overall health was suggested
Yang, 2015 [39]	Cross- Sectional (City)	999 (534)	45–60 Study (51.06 ± 4.45, *n* = 345), Control (50.92 ± 4.76, *n* = 654)	NAFLD (MetS)	Clinical and laboratory examination	Validated FFQ (143)	Traditional Chinese, animal food, grain–vegetable, high-salt dietary patterns	Animal food dietary pattern was associated with an increased risk of NAFLD
Xu, 2016 [40]	Longitudinal (Nation, 7)	1086 (568)	≥60	Obesity (MetS)	Anthropometric measurements	Three-day 24 h dietary recalls	Traditional, modern patterns	Rice-based traditional dietary pattern led to lower weight
He, 2017 [41]	Cross- Sectional (City)	1204 (461)	45–59 Study (50.79 ± 4.65, *n* = 243), Control (51.21 ± 4.69, *n* = 961)	Hyperuricemia (MetS)	Medical records	Validated FFQ and the Kaiser–Meyer–Olkin Measure	Traditional Chinese, meat food, mixed food patterns	The traditional Chinese dietary pattern was associated with a decreased risk of hyperuricemia, whereas the meat food pattern was associated with an elevated risk.
Shu, 2017 [42]	Cross- Sectional (City)	1918 (922)	45–59	T2D (MetS)	Medical records and histopathological data, anthropometric measurements	Validated FFQ (30)	Traditional Southern Chinese, western, grain–vegetable patterns	The Western dietary pattern was associated with increased risk of T2D, whereas the grain–vegetable dietary pattern was associated with reduced risk
Talaei, 2017 [43]	Longitudinal (Nation, 10.9)	54,341 (35,303)	45–74 (55.2 ± 7.6)	T2D (MetS)	Self-reported	Semiquantitative FFQ (165)	Not mentioned	Replacement of red meat and poultry with fish/shellfish reduced T2D risk
Liu, 2018 [44]	Cross- Sectional (Nation)	2552 (1276)	≥75 (79.42)	Malnutrition (MetS)	Face-to-face interview with validated questionnaire	Three-day 24 h dietary records	Rice staple, wheat staple, animal oil, plant oil diet patterns	Older adults were recommended to improve the quality of their diet by consuming various food sources, including good- quality fats and adequate vegetables
Wei, 2018 [45]	Cross- Sectional (City)	1918 (922)	45–59	Metabolic Syndrome (MetS)	International Diabetes Federation criteria	Validated FFQ and the Kaiser–Meyer–Olkin measure (138)	Traditional Chinese, animal food, high-energy patterns	The traditional Chinese dietary pattern was associated with reduced risk of MetS, whereas the animal food pattern was associated with a greater risk
Zhao, 2019 [46]	Cross- Sectional (Province)	688 (400)	60–82 (67.6 ± 4.2)	Gastrointestinal Discomfort (MetS)	The gastrointestinal symptom rating scale	Validated semiquantitative FFQ and 24 h dietary recalls (14)	Salt–tea, tuber–fruit–aquatic product–soybean, cereal–vegetable–meat diet patterns	Gastrointestinal discomforts were associated with food choice rather than dietary pattern or nutrient intake
Shen, 2020 [47]	Cross- Sectional (Province)	1761 (847)	45–59 Study (50.67 ± 4.52, *n* = 305), Control (52.82 ± 5.76, *n* = 1456)	Prediabetes (MetS)	Laboratory measurements	The Kaiser–Meyer– Olkin measure and the Bartlett test of sphericity	Traditional Southern Chinese, western, grain–vegetable patterns	Increased grain–vegetable and decreased western diet patterns were important in the early prevention of prediabetes
Xu, 2020 [48]	Cross- Sectional (City)	5281 (2755)	≥60 (67:90 ± 6:62)	T2D (MetS)	Clinical and laboratory examination	Questionnaire interview	Not mentioned	Long-term tea consumption was an independent protective factor for diabetic retinopathy
Chen, 2021 [49]	Cross- Sectional (City)	3795 (1912)	≥60	Sarcopenic Obesity (MetS)	Medical records and anthropometric measurements	Validated FFQ, 22	Meat–fish, lactoovovegetarian, junk food patterns	Lactoovovegetarian dietary pattern was inversely associated with the risk of SO in the elderly
Lee, 2021 [50]	Longitudinal (Nation, 2)	3253 (1702)	≥60	Obesity (MetS)	Anthropometric measurements	Self-reported	Not mentioned	Increased staple food intake was correlated with higher obesity risk
Chen, 2022 [51]	Longitudinal (Nation, 6)	1877 (1126)	50–70 Study (58.5 ± 5.98, *n* = 499), Control (58.1 ± 6.01, *n* = 1378)	T2D (MetS)	Laboratory measurements	Validated FFQ, 18 (74)	Not mentioned	Choosing the right dietary pattern based on the food metabolites was suggested
Song, 2022 [52]	Cross- Sectional (Nation)	40,909 (21,196)	≥45	Metabolic Syndrome (MetS)	National Cholesterol Education Program Expert Panel on Detection	Three-day 24 h dietary recalls, 23	Diverse, northern, southern patterns	Diverse pattern decreased the risk of central obesity
Yang, 2022 [53]	Cross- Sectional (Nation)	18,691 (9359)	≥60 (66.51)	Hyperuricemia (MetS)	Clinical and laboratory examination	Validated FFQ, 27 (64)	Typical Chinese, modern Chinese, tuber and fermented vegetable patterns	Effective dietary intervention could prevent whole noncommunicable chronic diseases among elderly people and assist in achieving healthy aging
Liu, 2017 [54]	Longitudinal (Province, 11.1)	4000 (2000)	≥65	Cancer Mortality (CA)	International Classification of Disease Version 10 codes	Validated FFQ, 15 (150)	Not mentioned	Cooking methods could influence the food components and the exposure of acrylamide
Wang, 2017 [55]	Longitudinal (Nation, 6.9)	63,257 (34,028)	45–74 (56.4 ± 7.7)	Gastric Adenocarcinoma (CA)	The Singapore Cancer Registry under the National Registry of Diseases Office	Validated FFQ	Vegetable–fruit–soy, meat–dim sum patterns	Vegetable–fruit–soy was healthier; alcohol was not recommended
Dong, 2015 [56]	Cross- Sectional (Province)	894 (648)	55–76 (62.93)	Cognitive Impairment (CIMI)	Montreal Cognitive Assessment	Validated FFQ, 13 (41)	Nut- and vegetable-rich diet pattern	Nut- and vegetable-rich diet was beneficial for cognitive function
Yu, 2015 [57]	Cross- Sectional (Province)	1717 (849)	≥65 (71.18 ± 4.97)	Depression (CIMI)	Patient Health Questionnaire-9	Questionnaire by face-to-face interview	Not mentioned	Frequent consumption of soybeans and soybean products decreased the risk of depressive symptoms
Zhao, 2015 [58]	Case–Control (Province, 0.25)	404 (207)	≥60 Study (84.63 ± 3.12), control (71.20 ± 7.31)	MCI (CIMI)	Montreal Cognitive Assessment Test	Validated FFQ, 11	Not mentioned	Intake of adequate eggs and marine products reduced the risk of developing MCI
Yuan, 2016 [59]	Case–Control (Province, 0.25)	276	55–75 Study (64.71 ± 0.52, *n* = 138), control (64.23 ± 0.47, *n* = 138)	Cognitive Impairment (CIMI)	Montreal Cognitive Assessment Test	Validated semiquantitative FFQ (41)	Not mentioned	Diets with low amounts of fish and high amounts of red meat contributed to cognition impairment
Sun, 2018 [60]	Cross- Sectional (Province)	339 (196)	≥60 (70.8 ± 7.9)	Cognitive Function (CIMI)	The Mini-Mental State Examination	Validated FFQ, 9	Not mentioned	Enhancing dietary knowledge allowed individuals to adopt healthy diets
Wang, 2018 [61]	Cross- Sectional (City)	1360 (235 Case, 1125 Control)	45–59 Study (54.82 ± 4.61), Control (50.36 ± 4.56)	Depression (CIMI)	The Center for Epidemiologic Studies Depression Scale	Validated FFQ and the Kaiser–Meyer–Olkin measure	Traditional Chinese, western, grain–vegetable, high-salt patterns	Increasing grain–vegetable diet pattern intake and decreasing western diet pattern intake was suggested
Yin, 2018 [62]	Cross- Sectional (Province)	1504 (736)	≥60 Study (72.9 ± 7.7, *n* = 290), Control (67.8 ± 6.2, *n* = 1214)	Cognitive Function (CIMI)	The Mini-Mental State Examination	Validated FFQ and 24 h dietary recalls	Mushroom–vegetable–fruit, meat–soybean product patterns	Increasing intake of vegetables–fruits could improve cognitive function
Chen, 2019 [63]	Cross- Sectional (City)	1676 (890)	45–59 Study (54.7 ± 7.60), Control (52.1 ± 5.80)	Depression (CIMI)	Self-report the Center for Epidemiologic Studies Depression scale	Validated FFQ over the past 12 months	Not mentioned	Higher consumption of vegetables and fruits was significantly associated with a lower risk of depressive symptoms
Shi, 2019 [64]	Cross- Sectional (Nation)	4685 (2437)	≥55	Cognitive Impairment (CIMI)	Telephone interview for cognitive status	Three-day 24 h dietary interview	Iron-related dietary pattern	Adequate animal food was required to prevent cognition decline among those with a higher intake of plant-based diet
Wang, 2020 [65]	Longitudinal (Nation, 3)	5716 (3064)	≥45	Cognitive Impairment (CIMI)	The Mini-Mental State Examination	Self-reported by trained research assistants, 8	Not mentioned	Choosing the healthier diet pattern was suggested
Xu, 2020 [66]	Longitudinal (City, 1)	1360 (741)	45–59 Study (50.13 ± 4.81), Control (53.79 ± 4.25)	Depression (CIMI)	A Simplified Chinese version of the Generalized Anxiety Disorder Scale	Semiquantitative validated FFQ	Traditional Chinese, western, grain–vegetable, high-salt diet patterns	The Western pattern was associated with an increased risk of anxiety, whereas the grain–vegetable pattern was associated with a reduced risk
Ding, 2021 [67]	Cross-Sectional (City)	1176 (659)	65–85 Study (73.5 ± 5.7, *n* = 284), Control (71.2 ± 5.1, *n* = 892)	Cognitive Impairment (CIMI)	The Mini-Mental State Examination	Validated FFQ, 15 (97)	Healthy, multigrain, snack dietary patterns	A well-structured and varied diet could protect older adults from declines in cognitive function
Duan, 2021 [68]	Cross- Sectional (Province)	3111 (1692)	≥60 Study (68.37 ± 5.53, *n* = 326), Control (67.55 ± 4.74, *n* = 2785)	MCI (CIMI)	Clinical examinations and the Mini-Mental State Examination	Questionnaire based on dietary guidelines for Chinese residents (2016)	Not mentioned	Eating breakfast frequently, drinking water before breakfast, consuming more water, and having lunch later (after noon) were negatively associated with the risk of MCI
Fu, 2021 [69]	Cross- Sectional (Province)	4457 (2471)	≥60	MCI (CIMI)	The Mini-Mental State Examination	Validated FFQ, 19	Methionine cycle metabolite-related, vegetarian, processed food diet patterns	Adopting MCM-related dietary patterns, especially avoiding processed foods, decreased the occurrence of MCI
Huang, 2021 [70]	Cross- Sectional (Nation)	4309 (2353)	55–86 (68.4)	MCI (CIMI)	The Montreal Cognitive Assessment, the Memory Index Score, and other scores	Validated FFQ, 13 (81)	Not mentioned	Higher consumption of rice, legumes, fresh vegetables, fresh fruit, meat, and nuts was primarily considered health
Jin, 2021 [71]	Cross- Sectional (Nation)	6160 (3551)	≥80 (90.1 ± 7.2)	Cognitive Impairment (CIMI)	The Mini-Mental State Examination	Face-to-face interviews by trained interviewers, 6	Not mentioned	Maintaining a healthy lifestyle, including dietary habits, throughout the life cycle course was suggested
Shang, 2021 [72]	Longitudinal (Nation, 7)	2307 (1172)	55–89 (63.3 ± 7.0)	Cognitive Decline (CIMI)	Telephone interview for cognitive status	Three-day 24 h dietary recalls	Dairy–fruit–fast food, grain–vegetable–pork, plant-based food, bean–mushroom, beverage–nut patterns	Bean–mushroom was a healthy pattern for brain health
Yeung, 2021 [73]	Longitudinal (Province, 4)	1518 (496)	≥65	Cognitive Impairment (CIMI)	Validated cognitive score of the Chinese version of the Community Screening Instrument of Dementia and the Mini-Mental State Examination	Validated FFQ (280)	Vegetable–fruit pattern	Increasing variety in vegetable–fruit intake could prevent the onset of cognitive impairment
H, 2022 [74]	Cross- Sectional (Province)	5410 (2953)	≥60 (71.04 ± 7.10)	Cognitive Impairment (CIMI)	The Mini-Mental State Examination	Face-to-face interview	Vegetable–fruit pattern	Daily vegetable and fruit intake was beneficial for cognition maintenance
Li, 2022 [75]	Longitudinal (Province, 6)	9028 (4587)	≥60 (68.7 ± 7.0)	Cognitive Impairment (CIMI)	The Mini-Mental State Examination	Questionnaire of baseline survey	Not mentioned	Limited egg consumption (less than 6 eggs/week) was prospectively related to a lower risk of cognitive impairment
Li, 2022 [76]	Cross- Sectional (City)	288 (152)	68–79 (72.0)	Mental Disorders (CIMI)	The Generalized Anxiety Disorder Scale	Face-to-face interviews and questionnaires	Dietary Diversity Score pattern	Eating a wide variety of foods was suggested
Liang, 2022 [77]	Longitudinal (Nation, 7)	4792 (2367)	≥65 (80.70 ± 9.58)	Cognitive Impairment (CIMI)	The Mini-Mental State Examination	Validated FFQ, 16	Overall, healthful, and unhealthful plant-based diet patterns	Higher unhealthful plant-based diet was associated with an increased risk of cognitive impairment
Zhu, 2022 [78]	Longitudinal (Nation, 10)	6136 (2843)	≥65 (80 ± 9.83)	Cognitive Function (CIMI)	The Mini-Mental State Examination	Validated FFQ, 16	Overall, healthful, and unhealthful plant-based diet indices	A diet rich in healthy plant foods with some healthy animal foods, such as fish, could benefit cognitive function
Qi, 2023 [79]	Cross- Sectional (Nation)	11,623 (6197)	≥65 (83.21 ± 10.98)	Depression (CIMI)	The 10-item Center for Epidemiologic Studies Depression Scale	Simplified FFQ	Healthful plant-based, unhealthful plant-based, animal-based diet patterns	Healthful plant-based foods were associated with a lower prevalence of depression and anxiety
Qin, 2023 [80]	Longitudinal (Nation, 10)	2454 (2293)	≥65 (75.46 ± 8.21)	Cognitive Impairment (CIMI)	Chinese version of the Mini-Mental State Examination	Face-to-face interviews by trained Interviewers, 2	Vegetable–fruit pattern	Frequent consumption of both fruits and vegetables led to a reduction in MCI risk
Chen, 2024 [81]	Cross- Sectional (Nation)	14,318 (7883)	≥65 (79.13)	Cognitive Impairment (CIMI)	The Mini-Mental State Examination	Validated FFQ, 24	Dietary Diversity Score pattern	Maintaining high dietary diversity could reduce the risk of cognitive impairment
Wang, 2024 [82]	Cross- Sectional (Nation)	8692 (4796)	≥60 (83.53 ± 11.48)	Cognitive Impairment (CIMI)	The Mini-Mental State Examination	Validated FFQ	Protein-enriched food, anti-inflammatory food patterns	Intake of more protein-enriched and anti-inflammatory food was suggested
Wang, 2024 [83]	Cross- Sectional (Nation)	11,971 (6401)	≥60 (83.23 ± 11.10)	Depression (CIMI)	The Center for Epidemiologic Studies Depression Scale-10	Simplified FFQ	Plant-based, healthful plant-based, unhealthful plant-based diet index patterns	Plant-based diet was negatively associated with depression and anxiety symptoms
Yang, 2024 [84]	Cross- Sectional (Nation)	14,150 (7937)	≥65 (85.33 ± 11.55)	Cognitive Impairment (CIMI)	The Mini-Mental State Examination	Validated FFQ	Not mentioned	Edible mushrooms and algae decreased the risk of cognitive impairment
Odegaard, 2014 [85]	Longitudinal (Nation, 5)	52584	45–74	Mortality (PSQ)	Nationwide registries of birth and death	24 h dietary recall interview and validated FFQ (165)	Vegetable–fruit–soy-rich, dim sum–meat-rich dietary patterns	Higher intake of plant-based foods, such as vegetables, fruits, soy and other legumes, whole grains, and nuts and seeds were associated with increased longevity
Liu, 2015 [86]	Cross- Sectional (Province)	3995 (1997)	≥65 (72.5 ± 5.2)	Bone Mineral Density (PSQ)	Clinical and laboratory examination	Validated FFQ (92)	Not mentioned	Greater fruit intake was associated with better bone mineral status
Shi, 2015 [87]	Longitudinal (Nation, 4.3)	8959 (5392)	≥80 (90.1 ± 6.9 for Male, 93.8 ± 7.7 for Female)	Mortality (PSQ)	Interviews with a close family member	Self-reported in face-to-face interviews	Not mentioned	Controlling intake of staple foods and increasing intake of fruits and vegetables were inversely associated with all-cause mortality
Xu, 2015 [88]	Cross- Sectional (Nation)	2401 (1264)	≥60	Anemia (PSQ)	Hospital records	Three-day 24 h dietary recalls, 27	Traditional, modern patterns	Overall healthier diet, rather than iron supplementation, was beneficial for preventing risk of anemia
Chen, 2016 [89]	Case–Control (City, 2)	210	50–65 (56.4 ± 3.8)	Bone Mineral Density (PSQ)	Hospital records	Three-day food records and validated FFQ	Not mentioned	Sufficient calcium dietary supplementation from food was suggested
Zhang, 2017 [90]	Case–Control (Province, 5)	2010 (1562)	52–83 Case and Control: 1050	Hip Fracture (PSQ)	Hospital records	Self-reports and Dietary Inflammatory Index (DII) score calculation	Proinflammatory, anti-inflammatory diet patterns	Increasing intake of the anti-inflammatory diet and decreasing the intake of the proinflammatory diet were suggested
Zhang, 2018 [91]	Cross- Sectional (Province)	738 (336)	55–77 (61.6 ± 8.3)	Anemia (PSQ)	Hemoglobin concentration was determined by the cyanmethemoglobin method by local laboratory technicians	24 h recalls and household food inventory	Not mentioned	A balanced diet with more vegetables, fruits, dairy, soybean, eggs, and fish and less cereals, meat, cooking oil, and salt was encouraged
Lv, 2019 [92]	Longitudinal (Nation, 3.4)	28,790 (17,779)	>80 (92.9 ± 7.5)	Mortality (PSQ)	From close family members or doctors	Validated FFQ	Dietary Diversity Score pattern	Higher Dietary Diversity Score was associated with significantly lower mortality risk
Li, 2020 [93]	Cross- Sectional (City)	861 (456)	≥65 (71.0 ± 4.8)	Sarcopenia (PSQ)	Asian Working Group for Sarcopenia in 2014	Validated FFQ, 21	Mushrooms–fruit–milk oil, mushrooms-fruit, animal foods patterns	Low PEF (<30%) was negatively associated with sarcopenia
Tao, 2020 [94]	Longitudinal (Nation, 3.2)	17,949 (10,362)	≥65 (87.39 ± 11.53)	Mortality (PSQ)	Interviews with close family members or the doctor	Self-report, 9	Dietary Diversity Score pattern	Dietary diversity was inversely associated with all-cause mortality
Aihemaitijiang, 2022 [95]	Longitudinal (Nation, 7)	2282 (1123)	≥60	Physical Function (PSQ)	Anthropometric measurements, Katz Index, and IADL Scale	Validated FFQ, 13	Fruit–egg–milk, vegetable–meat–fish, condiment–tea patterns	Long-term maintenance of high dietary diversity could help maintain good physical function in the Chinese elderly population
Shen, 2022 [96]	Longitudinal (Nation, 5.7)	13,156 (7552)	≥65 (86.9 ± 11.4)	Mortality (PSQ)	Official death certificate or questionnaire responses by close relatives	Validated FFQ and face-to-face Interview	Not mentioned	Regular consumption of mushrooms and algae during mid to late life demonstrated the strongest benefit against death
Yan, 2022 [97]	Longitudinal (Nation, 5.2)	35,927 (22,008)	80–124 (92.0)	Mortality (PSQ)	Collected from officially issued death certificates	Validated FFQ, 9	Not mentioned	Poorer diet was associated with mortality from any cause among Chinese older adults
Zhang, 2022 [98]	Longitudinal (Nation, 4)	2047 (988)	≥60	Frailty (PSQ)	Frailty index assessment	Validated FFQ, 13	Milk–nut–mushroom/algae, egg–bean–pickle–sugar, fruit–vegetable–meat–fish, tea patterns	Early initiation and long-term Maintenance of a diversified diet should be encouraged to reduce the risk of frailty
Bian, 2023 [99]	Cross- Sectional (Province)	515 (312)	≥65 (71.31 ± 4.71)	Sarcopenia (PSQ)	Asia Working Group on Sarcopenia 2019 consensus	Validated FFQ (70)	Dietary Inflammatory Index pattern	Inadequate dietary nutrient intake was an urgent problem to address in the Chinese elderly population
Dai, 2024 [100]	Longitudinal (Nation, 3)	4838 (2405)	≥65 (80.8 ± 9.6)	Frailty (PSQ)	Frailty index assessment	Self-report and questionnaire	Not mentioned	Animal fat, especially dietary oil from fish, was associated with a significantly decreased risk of frailty
Gao, 2024 [101]	Longitudinal (Nation, 5.8)	7166 (3804)	≥65 (81.46 ± 10.18)	Frailty (PSQ)	Frailty index assessment	Validated FFQ, 22	Plant-based pattern	Plant-based dietary patterns attenuated the association between frailty and cognitive decline
Huang, 2024 [102]	Longitudinal (Nation, 9)	7843 (4255)	≥60 (82.2 ± 10.9)	Mortality (PSQ)	Reported from family members, neighbors and caregivers; the village doctor; and local residential committees	Validated FFQ	Plant-based, healthful plant-based, unhealthful plant-based diet index patterns	Healthful plant-based diets were suggested
Sun, 2024 [103]	Longitudinal (Province, 15.8)	19,598 (14,023)	≥50 (62.7 ± 6.7)	Mortality (PSQ)	Death registry of the Guangzhou Center for Disease Control and Prevention	Validated FFQ	Vegetable-based, healthy Cantonese, western, nut–fruit dietary patterns	The healthy Cantonese and nut–fruit dietary patterns were suggested

^a^ The sampling was from city, province, or nation, which represented that the sampling level was municipal, provincial, or national, respectively. The numbers within parentheses represent follow-up years. ^b^ The numbers within parentheses represent mean age at baseline. ^c^ N: number of food groups involved; n: number of food items involved. Abbreviations: FFQs: food frequency questionnaires; BP: blood pressure; DASH: dietary approaches to stop hypertension; cMIND: Mediterranean–DASH intervention for neurodegenerative delay; CCVDs: cardiocerebrovascular diseases; NAFLD: nonalcoholic fatty liver disease; T2D: type 2 diabetes mellitus; IADL: instrumental activities of daily living; SO: sarcopenic obesity; MCM: methionine cycle metabolites; PEF: percentage of energy from fat.

## Data Availability

No new data were created or analyzed in this study. Data sharing is not applicable to this article.

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
