# Peer review of "Uncovering a New Concept of Foodnality in Diet Recommendations for Chinese Aging Populations Based on Systematic and Bibliometric Review"

_foods, 2024, doi:10.3390/foods13244062_

Round 1

Reviewer 1 Report

Comments and Suggestions for Authors

Dears, 

The present manuscript proposes a new concept of dietary recommendations for elderly Chinese populations based on three systematic and bibliometric reviews. As suggested in the title, this is a comprehensive bibliometric analysis of population-based research from China that has been systematically examined over the last 10 years. Cross-sectional, longitudinal and case-control studies were conducted to investigate the correlations among foods, dietary patterns and healthy aging conception. I consider that the methodology is quite adequate and well described, as are the analyses and results. Figure 1 represents very well the interaction between diet and human health through three main pathways, however, figure 4 is very difficult to read and could be simplified, given its relevance. As it is a very important journal in the area, a more comprehensive discussion and conclusion is expected, including comparison with results from international studies relevant to the topic. The references used are predominantly restricted to China, limiting the scope of the work

Author Response

The present manuscript proposes a new concept of dietary recommendations for

elderly Chinese populations based on three systematic and bibliometric reviews. As

suggested in the title, this is a comprehensive bibliometric analysis of population-based research from China that has been systematically examined over the last 10years. Cross-sectional, longitudinal and case-control studies were conducted to investigate the correlations among foods, dietary patterns and healthy aging conception. I consider that the methodology is quite adequate and well described, as are the analyses and results. Figure 1 represents very well the interaction between diet and human health through three main pathways, however, figure 4 is very difficult to read and could be simplified, given its relevance.

Response

Thank you for acknowledging our manuscript and providing valuable suggestions.

Figure 4 is reflecting clustering analysis of recommended food items (a) and unrecommended food items (b) in association with chronic disease symptoms based on bimodal matrix. The bimodal matrix diagram represented the relationship between food types (horizontal axis) and the physical functions of chronic diseases (vertical axis). Each cell in the matrix displayed a frequency number, reflecting the degree of association. At above and left of Figure 4a and 4b, there are connection lines indicating high potential correlation combinations of elements in horizontal and vertical axis respectively. However, the primary focus of this diagram is to represent the correlation between food and chronic disease symptoms, and it does not involve any discussion about intra-group element. Therefore, we removed this part to simplify the image, as shown below. We hope you can accept our adjustments, and we greatly appreciate your understanding.

Figure 4: Lines 259-264

As it is a very important journal in the area, a more comprehensive discussion and conclusion is expected, including comparison with results from international studies relevant to the topic.

Response

Thank you for bringing this to our attention, and we appreciate your careful review

of our manuscript. International cohort studies on food primarily focus on dietary patterns, which also reflected in the cohort studies included in our literature search. However, the purpose of this paper is to highlight the importance of food attributes or foodnality in constructing healthy dietary patterns. By gaining a deeper understanding of food attributes or foodnality, we can make more precise food choices, thereby enhancing the quality of overall dietary patterns. To illustrate this point, we make comparison with international dietary patterns and explain how to apply foodnality to develop a premium dietary pattern for Chinese aging populations:

Lines 499-509:

For example, the Mediterranean diet emphasizes whole grains, and olive oil as sources of

energy, but it’s more reasonable to suggest Chinese aging populations to consume more coarse cereals, tubers, tea oil or linseed oil in order to enjoy the similar positive foodnality. In the DASH diet, the high intake of fruits and vegetables aligns with the plant-based foods in traditional Chinese diets, however, milk is not suitable for Chinese individuals with lactose intolerance. Chinese aging populations may need to seek lactose-free dairy products to shift the foodnality in the DASH diet to enhance the dietary quality. The absence of fish in vegetarianism may not suit Chinese elderly individuals who require source of high-quality protein, therefore, it is recommended that vegetarian seniors undergo dietary assessments and supplement with some animal foods with positive foodnality.

The references used are predominantly restricted to China, limiting the scope of the work

Response

Thank you for this valuable comment. Considering the theme of this manuscript is

to assist the aging population in China in cultivating healthy dietary concept and selecting personalized foods that are more suitable for their individual physical status, we have incorporated Medical Subject Headings (MeSH) ‘China’, into our screening criteria. Then, we have fully reviewed 332 articles to make sure the cohort studies are related Chinese aging population and meet the screening criteria. This has resulted in the final 83 studies included in our review focusing on Chinese in China and Singapore. Although this approach limits the scope of our research, the relationships between the foods discussed in the articles and the aging populations in China are more compelling. We have added 3 more cohort study references from Europe as below, but still not that many. We acknowledge your suggestions which really give us inspiration for future research to expand internationally perspective to validate the broader potential applications of foodnality.

  1. Smith, R.; Methven, L.; Clegg, M. E.; Geny, A.; Ueland, Ø.; Grini, I. S.; Rognså, G.H.; Maitre, I.; Brasse, C.; Wymelbeke-Delannoy, V.V.; et al. Older adults' acceptability of and preferences for food-based protein fortification in the UK, France and Norway. Appetite 2024, 197, 107319.
  2. Feraco, A.; Armani, A.; Amoah, I.; Guseva, E.; Camajani, E.; Gorini, S.; Strollo, R.; Padua, E.; Caprio, M.; Lombardo, M. Assessing gender differences in food preferences and physical activity: a population-based survey. Front. Nutr. 2024, 11, 1348456.
  3. Daniele, G. M.; Medoro, C.; Lippi, N.; Cianciabella, M.; Magli, M.; Predieri, S.; Versari, G.; Volpe, R.; Gatti, E. Exploring Eating Habits, Healthy Food Awareness, and Inclination toward Functional Foods of Italian Elderly People through Computer-Assisted Telephone Interviews (CATIs). Nutrients 2024, 16, 762.

We have tried our best to improve the manuscript and made some changes in the manuscript. These changes will not influence the content and framework of the paper.

We appreciate for Editors and Reviewers’ warm work earnestly, and hope that the correction will meet with approval.

Once again, thank you very much for your comments and suggestions.

Reviewer 2 Report

Comments and Suggestions for Authors

Congratulations to the authors for their work. Attached are some proposed changes to improve the presentation of the research. 

Author Response

STRUCTURE

The manuscript is correctly structured. But there are some points to consider:

  1. Authors are encouraged to review Figure 1. They should be inserted in the main text near their first citation. To facilitate the editing of larger tables, smaller fonts may be used, but no smaller than 8 point.

Response

Thank the reviewer for the comment. We have adjusted the position of Figure 1 to

bring it closer to its first citation location. Additionally, we have reviewed other images and tables to ensure they are also placed correctly. And we have adjusted the table font size to 8 point, due to space constraints, we have opted not to list them here but have marked them in red in the manuscript.

   We list the line numbers of Figures and first citation as below:

Figure 1: Lines 45-52, citation: Line 53

Figure 2: Lines 95-108, citation: Line 92

Figure 3: Lines 243-248, citation: Line 231

Figure 4: Lines 259-264, citation: Line 258

Figure 5: Lines 266-270, citation: Line 257

Figure 6: Lines 289-292, citation: Line 273

Table 1: Lines 176-129, citation: Line 166

  1. The same for figure 2, the format has to match the one indicated in the magazine. Applicable to the rest of the manuscript.

Response

Thank you for your suggestion. We have adjusted the format of the figure 2 to

ensure uniformity. The font used is Times New Roman, with the main context font at size 8, except for the main titles. (Figure 2: Lines 95-108)

  1. Table 1. When a table is split into two sheets, the sections must be put back in the header. Applicable to the rest of the manuscript.

Response

Thank you for your kind reminding. Due to the systematic summarization of

screened literatures, the Table 1 spans multiple pages. Therefore, following your suggestion, we have split the table and added “Table 1 Cont.” at the table header of each page to clearly indicate that it is a continuation of the same table. Due to space constraints, we have opted not to list them here but have marked them in red in the manuscript. (Lines 176-129; Pages 5-19)

  1. Line 379: “resulting in negative foodnality in our eligible database”. Remember not to speak in the first-person plural.

Response

Thank you for your valuable comments. We have deleted ‘our’ in the manuscript,

and checked the rest content to make sure not speak in the first-person plural. We’ll pay careful attention for this requirement.

TITLE AND ABSTRACT

The title mentions the type of research that has been carried out. The summary is well structured, listing all parts.

  1. Remember: Abstract: The abstract should be a total of about 200 words

Response

Thank you for your recognition and valuable comments. We have removed the

following portions from the original abstract, with no impact on the overall meaning of the abstract. After checking the word count, it meets the requirements of about 200 words.

Line 10: Healthy dietary intervention is essential to prevent chronic diseases and to extend life for fostering healthy aging. (delete)

  1. https://www.mdpi.com/journal/foods/instructions#figuresLine

Response

Thank you for your kind reminding. Following your instruction, we have reviewed

the manuscript guidelines provided by the journal and verified the formatting and content of the manuscript.

INTRODUCTION

Some sections could be better connected to improve the flow of ideas. For example, the transition between the paragraph on ageing and the paragraph on diet is not entirely smooth:

  1. Line 42: “Promoting healthy aging was carried out as an essential national policy of the ‘14th Five-Year Plan’ to maintain social and economic stability [8]. Diet is intrinsically related to nutritional support, chronic disease management and healthy ageing, prompting extensive study of the relationship between diet and overall health index through population monitoring [9]”.

Response

Thank you for your recognition and valuable comments. We feel that your

description has enhanced the logical flow of the language. Accordingly, we have made modifications to the original content based on your suggestion:

Lines 40-45:

The promotion of healthy aging has been implemented as a key national policy within the ‘14th Five-Year Plan’, with the aim of maintaining social and economic stability [8]. In this context, diet plays a key role in nutritional support, chronic disease management, and the promotion of healthy aging. This has prompted numerous studies on the relationship between diet and general health indicators through population monitoring [9].

It might be useful to explain that, due to the particularity of the Chinese diet and

the needs of the older population, adapting Western patterns such as the Mediterranean diet or DASH may not be appropriate. This may lead to further analysis on foodnality (diet personalisation), which is the central theme of the study:

  1. Line 66: “However, those dietary patterns may not be suitable for ageing Chinese populations”.

Response

Thank you for your valuable comments. Your description is extremely precise, and

we have adopted your suggestions to revise the content as below:

Lines 72-74:

However, those dietary patterns may not be suitable for the Chinese aging populations due to specific cultural and metabolic differences.

MATERIAL AND METHODS

The first paragraph is somewhat dense and contains technical terms that could be better explained. Recommendation: break the paragraph into shorter sentences and clarify the objectives.

  1. Line 90: “The systematic and bibliometric review followed the 2020 Preferred Reporting Items for Systematic Reviews and Meta-Analyses (PRISMA) guidelines [19], and … and co-occurrence analysis”.

Response

Thank you for your professional suggestion. We have referred to your

description because it makes the article more concise:

Lines 90-95:

This systematic and bibliometric review followed the PRISMA 2020 (Preferred Reporting Items

for Systematic Reviews and Meta-Analyses) guidelines [19]. In the process, the PRISMA procedure was used to select relevant articles, as summarized in Figure 2. Scientific mapping techniques, such as co-word and co-occurrence analysis, were then applied to identify exposed foods and dietary patterns, chronic disease outcomes and dietary recommendations.

Study design. The research presents the key elements of the study design at the

beginning of this section.

  1. In addition, it is also important to specify the selection process of the studies, for example the screening and eligibility included in the systematic review.

Response

Thank you for your recognition and valuable comments. We have summarized the

selection process of this studies in Figure 2. In the manuscript, below content were provided for screening and eligibility included in the systematic review:

Screening (Lines 127-134):

Web of Science (WOS) all databases, including WOS core databases, BIOSIS Citation Index,

Chinese Science Citation Database, Korean Journal Database, MEDLINE, Preprint Citation Index, SciELO Citation Index, were conducted to search studies published between 1st June 2014 and 31st May 2024. We used ‘diet’ as the topic word, abstract word, or author keyword in the search strategies separately, and used Medical Subject Headings (MeSH) including ‘China’, ‘aged’, ‘middle aged’, ‘aging’, or ‘aged 80 and over” for further screening. In addition, further citation tracing searches were performed to ensure the research comprehensiveness to retrieve studies missed by predefined search.

Selection (Lines 136-145):

Food items, food groups, dietary patterns and diet recommendations were evaluated without limitation or restriction imposed on the geographical location, gender, lifestyle or income status of participants. Abstracts, protocols, conclusions, and full text was pre-viewed to exclude studies that did not meet the criteria: 1) be of cross-sectional cohort, lon-gitudinal cohort, or nested case-control design, including follow-up studies of randomized controlled trials; 2) participants aged ≥ 45; 3) provided recommendation or unrecommen-dation on the association with the chronic diseases and food items, food groups or dietary patterns. When exposure was dietary patterns, food items of factor loading ≥ 0.2 were screened; 4) specific food items must be contained, but excluded studies targeted on micro- and macronutrients.

RESULTS

It is recommended that the general results section be organised by topic, this makes it easier to read and understand, helping to quickly identify the effects of diet on different health conditions (cardiovascular disease, metabolic syndrome, etc.).

  1. Restructure into thematic subheadings: CVD, MetS, CA, CIMI, PSQ, Foodnality and Guidelines..

Response

Thank you for your recognition and valuable comments. Please allow us to explain

the logical structure of the content in the results section. First, based on our screen criteria, we ultimately screened 83 relevant articles, we aimed to summarize these articles, including the research types, primary findings, and the foods involved. Consequently, we have structured Section 3.1 and Table 1 accordingly. Most importantly, we conducted a co-occurrence analysis of the foods mentioned in the articles, aiming to identify which foods were recommended and which were not for the elderly in China, thus resulting in the content under Section 3.2. Furthermore, we analyzed the associations between these foods and five chronic diseases (CVD, MetS, CA, CIMI, PSQ), which was closely related health aging. We discovered that some recommended foods might yield opposite health effects in different physical status or processing conditions. Therefore, we introduced this concept ‘foodnality’ under Section 3.4. Finally, based on the primary dietary recommendations from the selected articles, we conducted a thematic analysis and proposed a theoretical framework for dietary recommendations. Therefore, we sincerely hope that you can accept this structure logic with our deepest appreciation.

DISCUSSION

To enhance the discussion of the study, the following points could be added:

  1. Clarification of the concept of ‘foodnality’ and its practical application

The definition of foodnality is presented conceptually, but without specific examples illustrating how this concept translates into practical recommendations. It is recommended to add specific examples on how foodnality is used to select foods, differentiating between foods with high positivity and those with dual effects depending on their preparation.

Response

We appreciate the reviewer’s professional comment. We found there were many

factors to influence the foodnality in practical application scenarios, including food quality, food diversity, food processing, aging people’s physical status. So we suggest to analyze the foodnality first and then to make diet choice, such as evaluate the freshness of foods, diet with high foods diversity, and evaluate individual health status. Based on these recommendations, we provided examples in the manuscript as below:

Lines 474-481:

Basically, consumers should evaluate the importance of the quality, diversity, and collocation related to foods. Theoretically, the priority is targeting at high-quality food items, namely fresh vegetables, fruits, fish, nuts, whole grains, rich nutrients and simplified processing for elderly people [25,28,44]. Increasing dietary diversity can ensure various foods intake to promote health on CIMI and PSQ, benefits for both the physical and mental well-being of the elderly [76,81,94,95]. In the CLHLS cohort study, the dietary diversity score was calculated according to the intake frequency of 13 food groups, and avoided the low diversity score was defined as <7 [92].

Lines 483-488:

In addition, food items would have coordinating or resistance effect in a diet, so that the elderly people are recommended to screen the food items according to their own physical status. For example, a traditional Chinese plant-based diet characterized by more intake of nuts, seeds, tubers, vegetables, legumes, fruits were beneficial for obesity. However, this diet might display negative effects on anemia, suggesting professional health assessment and dietary patterns adjustments are necessary when judging foodnality [69].

  1. Specificity in the recommendations for critical nutrient intakes

Proteins, carbohydrates and fats are mentioned in general terms, without suggestions for specific amounts or how to adjust intake according to the individual's health status. Provide specific recommendations on the intake of critical nutrients, such as limiting carbohydrate or protein, according to the risk of chronic diseases prevalent in older adults, such as metabolic syndrome or sarcopenia.

Response

Thank you for your valuable comments. We fully agree with your stance that from

precision nutrition perspective, it’s necessary to provide more accurate nutritional interventions and nutrient-based food selections, especially for aging populations with chronic disease. However, we did not extensively mention precise nutrient content and recommendations in this manuscript due to two primary reasons. Firstly, there was indeed a lack of substantial data in this area in our screened articles. From the exist researches, we found that the main types were cohort studies, which primarily conducted food frequency questionnaires, self report, call or facial review to evaluate food choices and dietary patterns among the elderly. Monitoring energy and nutrient intake in practical scenarios is challenging to implement. Secondly, we aimed to provide the elderly populations with a simplified food selection guideline in this manuscript because most older adults don’t have professional nutritional concept and it would be difficult for them to make food choices based on nutritional content in daily life.

Certainly, we believe that nutritional intervention is undoubtedly the trend of the future, and therefore we will continue to learn and make more research in this direction. In our manuscript, we have appended a background section on protein requirements to demonstrate the importance of adequate and balanced nutrient supplementation:

Lines 410-416:

For instance, Chinese Food Pagoda advocates older adults for increasing protein intake with a recommended range of 1.2–1.5 g/kg body weight to prevent muscle degradation, however, the objective fact is that the protein intake among elderly people is far from reaching this standard, even in European regions where animal meat diet is predominant [104]. It’s worth noting that excessive nutrients might bring burden to the elderly, so food items should be selected according to the individual status.

  1. Smith, R.; Methven, L.; Clegg, M. E.; Geny, A.; Ueland, Ø.; Grini, I. S.; Rognså, G.H.; Maitre, I.; Brasse, C.; Wymelbeke-Delannoy, V.V.; et al. Older adults' acceptability of and preferences for food-based protein fortification in the UK, France and Norway. Appetite 2024, 197, 107319.

  1. Integration of Foodnality into international dietary patterns

The discussion mentions patterns such as the Mediterranean diet but does not offer a practical adaptation to the Chinese context, nor does it suggest how to combine food with recognized patterns. Explore how to combine aspects of the Mediterranean or DASH diet with traditional Chinese ingredients and the concept of foodnality to develop hybrid and culturally relevant dietary patterns.

Response

Your suggestion is excellent. Due to the significant differences in food types and

dietary cultures between overseas countries and China, as well as the variations in physical status, it is not advisable to directly apply internationally dietary patterns to the elderly population in China. The intent of this paper is to optimize dietary quality through a deeper understanding of food characteristics. In term of the application of foodnality in international dietary patterns, we have made the following supplementary points:

Lines 499-509:

For example, the Mediterranean diet emphasizes whole grains, and olive oil as sources of energy, but it’s more reasonable to suggest Chinese aging populations to consume more coarse cereals, tubers, tea oil or linseed oil in order to enjoy the similar positive foodnality. In the DASH diet, the high intake of fruits and vegetables aligns with the plant-based foods in traditional Chinese diets, however, milk is not suitable for Chinese individuals with lactose intolerance. Chinese aging populations may need to seek lactose-free dairy products to shift the foodnality in the DASH diet to enhance the dietary quality. The absence of fish in vegetarianism may not suit Chinese elderly individuals who require source of high-quality protein, therefore, it is recommended that vegetarian seniors undergo dietary assessments and supplement with some animal foods with positive foodnality.

  1. Limitations and future research proposal

The research limitations mention variability in the studies, but do not address the need to investigate how public understanding of the concept of foodnality can be improved. Extend the limitations to point out the lack of research on educational strategies to help older adults apply the concept of foodnality in their daily lives. Suggest the development of specific and accessible guidelines for food choice based on food quality.

Adding these points to the discussion would provide a more comprehensive better contextualised analysis with clearer implications for public health policy and clinical practice.

Response

We sincerely appreciate the valuable comment. You are absolutely right, there are

many variable factors behind concept of ‘foodnality’, which is not easily understood by the elderly. Moreover, aging people sometimes hardly change the long-standing dietary habits, for example the porridge used to quickly replenish energy but now is easily increase their blood sugar amoing those aging populations who keep the habit of eating porridge in the morning. So our priority is to promote healthier eating habits to public. This also requires us to better disseminate foodnality and apply it in practice. Special thanks to you again for your good comments, which inspired us to follow up with related research. We have rewritten this part based on your suggestion as followings:

Lines 518-528:

The findings revealed foodnality is important in the association between diet and health. However, the interpretation of the new concept "Foodnality” is extremely complex, since substantial variations existed across the studies in terms of the definitions of outcomes, measurement of exposure, cohort design, and sample size. There is limited research on educational strategies to help older adults apply the concept of foodnality in their daily lives, implying the development of personalized and accessible guidelines for food choice based on food quality is necessary. Further research is needed on how to communicate the concept of foodnality in a clear and practical way to the older populations, so that they can make informed food choices. In addition, future studies could focus on developing rapid foodnality assessment tools that allow consumers to easily compare the nutritional quality and potential effects of different foods.

CONCLUSION

The conclusions section does appear, but it would be pertinent to rename some conclusions and limitations briefly and in line with aspects of improvement for future research. That is, including a final section of conclusions.

  1. g. ‘Further research is needed on how to communicate the concept of foodnality

in a clear and practical way to the older population, so that they can make informed food choices. In addition, future studies could focus on developing rapid foodnality assessment tools that allow consumers to easily compare the nutritional quality and potential effects of different foods.

Response

Thank you for your recognition and valuable comments. To enhance the clarity of

our conclusions, we have added a conclusion at the end of discussion as per your suggestion. But for the limitations and future research section, we have kept it at the end. Because we believe personalized nutrition for the elderly is just in beginning stage, more researches besides foodnality are required, especially against the backdrop of an aging population. The additional conclusion section is as follows:

Lines 512-516:

In conclusion, foodnality exists and it’s related to the characteristics and consumption scenarios

of foods. Foodnality is important for aging populations to better understand the dual character of foods based on the food quality, food processing, as well as individual physical status. Therefore, understanding the concept of foodnality enables the elderly to achieve a quicker selection of suitable food items according to their actual situation.

REFERENCES

  1. References are numbered in order of appearance in the text. They are correctly

placed in square brackets [ ] and placed before punctuation. In addition, most of the references are less than 5 years old.

Response

Thank you for your careful suggestion. We have checked our manuscript to make sure that the references follow the recommended format of the magazine. All references are matched with their corresponding numbers and appear in the correct position in the manuscript. Additionally, regarding our literature is a systematic review that incorporates cohort studies over the past decade, it is inevitable that we have cited literature from the years 2014-2024. We are appreciated you can understand this situation. Apart from these necessary cohort study references, all other references are sourced from the past five years (2019-2024) except one reference is in 2018 (No.17).

The newly added literature includes:

  1. Zhou, J.; Leepromrath, S.; Zhou, D. Dietary diversity indices v. dietary guideline-based indices and their associations with non-communicable diseases, overweight and energy intake: evidence from China. Public Health Nutr. 2023, 26, 911-933.
  2. Agnoli, C.; Baroni, L.; Bertini, I.; Ciappellano, S.; Fabbri, A.; Goggi, S.; Metro, D.; Papa, M.; Sbarbati, R.; Scarino, M.L.; et al. A comprehensive review of healthy effects of vegetarian diets. Nutr. Metab. Cardiovas. 2023,33, 1308-1315.
  3. Smith, R.; Methven, L.; Clegg, M. E.; Geny, A.; Ueland, Ø.; Grini, I. S.; Rognså, G.H.; Maitre, I.; Brasse, C.; Wymelbeke-Delannoy, V.V.; et al. Older adults' acceptability of and preferences for food-based protein fortification in the UK, France and Norway. Appetite 2024, 197, 107319.
  4. Feraco, A.; Armani, A.; Amoah, I.; Guseva, E.; Camajani, E.; Gorini, S.; Strollo, R.; Padua, E.; Caprio, M.; Lombardo, M. Assessing gender differences in food preferences and physical activity: a population-based survey. Front. Nutr.2024, 11, 1348456.
  5. Daniele, G. M.; Medoro, C.; Lippi, N.; Cianciabella, M.; Magli, M.; Predieri, S.; Versari, G.; Volpe, R.; Gatti, E. Exploring Eating Habits, Healthy Food Awareness, and Inclination toward Functional Foods of Italian Elderly People through Computer-Assisted Telephone Interviews (CATIs). Nutrients2024, 16, 762.
  6. Guo, Y.; Shi, S.; Yang, N.; Tang, M. X.; Duan, Z. J.; Guo, X. R.; Tang, Z. H. Comparative assessment of nutritional composition, polyphenol profile and antioxidative properties of wild edible ferns from northeastern China. Food Res. Int. 2023 163, 112237.

  1. Reference section follow the recommended style.

Response

Thank you for the crucial reminder. We have carefully reviewed our manuscript,

ensuring that the references follow the recommended style of the magazine.

We have tried our best to improve the manuscript and made some changes in the manuscript. These changes will not influence the content and framework of the paper.

We appreciate for Editors and Reviewers’ warm work earnestly, and hope that the correction will meet with approval.

Once again, thank you very much for your comments and suggestions.
